# A Review of Green Aerogel- and Xerogel-Based Electrodes for Supercapacitors

**DOI:** 10.3390/polym16192848

**Published:** 2024-10-09

**Authors:** Ngo Tran, Hyung Wook Choi, Quang Nhat Tran

**Affiliations:** 1Institute of Research and Development, Duy Tan University, Da Nang 550000, Vietnam; tranngo@duytan.edu.vn; 2Faculty of Natural Sciences, Duy Tan University, Da Nang 550000, Vietnam; 3Department of Electrical Engineering, Gachon University, 1342 Seongnamdaero, Sujeong-gu, Seongnam-si 13120, Gyeonggi-do, Republic of Korea; 4Department of Chemical and Biological Engineering, Gachon University, 1342 Seongnamdaero, Sujeong-gu, Seongnam-si 13120, Gyeonggi-do, Republic of Korea

**Keywords:** supercapacitors, aerogel, xerogel, pseudocapacitance, conductivity, flexibility

## Abstract

The decline in fossil fuels on the earth has become a primary global concern which has urged mankind to explore other viable alternatives. The exorbitant use of fuels by an ever-increasing global population demands a huge production of energy from renewable sources. Renewable energy sources like the sun, wind, and tides have been established as promising substitutes for fossil fuels. However, the availability of these renewable energy sources is dependent on weather and climatic conditions. Thus, this goal can only be achieved if the rate of energy production from renewable sources is enhanced under favorable weather conditions and can be stored using high energy storing devices for future utilization. The energy from renewable sources is principally stored in hydropower plants, superconducting magnetic energy storage systems, and batteries.

## 1. Introduction

The depletion of fossil fuel reserves on Earth has become a pressing global issue, prompting the exploration of alternative solutions. The ever-expanding global population’s extensive consumption of fuels necessitates a substantial increase in energy production from renewable sources. Renewable energy sources, including solar, wind, and tidal power, have emerged as promising alternatives to fossil fuels. However, their utilization is contingent upon favorable weather and climatic conditions. Achieving this goal requires enhanced energy production from renewable sources during favorable conditions, with surplus energy being stored using high-capacity storage devices for future utilization [1,2,3,4,5]. Renewable energy is primarily stored in hydropower plants, superconducting magnetic energy storage systems, and batteries.

Supercapacitors also termed as electrochemical capacitors, ultra-capacitors or electrical double layer capacitors (EDLC) are devices used for storing energy. They are attractive storage devices based on their electrochemical properties like high stability, power density, and fast charge–discharge capability [2,3]. These devices are being widely used in portable electronic devices and other heavy hybrid devices. They indeed offer energy storage capability and achieve charge accumulation through enhancing redox reactions, thereby facilitating rapid energy trapping and delivery. Regardless of these added advantages, supercapacitors are still limited for their application on a large scale, based on issues like stability and poor capacitance [4]. The materials constituting the electrode dictate the performance of the output of the capacitor. The charging speed and the storage capacity of electrical double layer capacitors are theoretically proportional to the surface area of the porous electrodes. Thus, there is a direct relation between parameters like pore structure and capacitance. Additionally, parameters like the size distribution of pores, pore volume, the size of particles, the composition of electrolytes and the functional groups present on the materials comprising the electrodes also affect the properties of the electrodes in supercapacitors [5]. The surface area of the materials comprising the electrodes should be very large. The capacity can also be increased by escalating the rate of the transfer of electrons [6,7]. An appropriate size of the pores along with suitable architecture and materials is the most important requirement for the fabrication of an efficient supercapacitor. 

Aerogels and xerogels are solid materials that are light in weight, possess a low density, and have a nano-sized structure, which are synthesized by the process of drying wet gels [8]. Liquid is substituted by air in aerogels by techniques that keep their volume intact, while the liquid phase present within a gel is eliminated by evaporation to form a xerogel. A comparison of the synthesis procedures of both aerogels and xerogels indicates that xerogels have a greater tendency to crack during their synthesis. Aerogels possess structured pores, a low density, and a highly branched structure, which is an added advantage for their use as capacitor materials [9,10]. The distribution of the pore size and surface area of aerogels can be manipulated by the modulation of different parameters depending on the nature of the solvent, reactants, and catalyst; the selection of appropriate drying techniques; and the identification of an effective route for synthesis along with the pyrolysis temperature [11]. It is significant to mention that in the case of carbon aerogels, the structure of carbon materials in the final aerogels is influenced by the structure of the organic compounds present as precursors [12]. Carbon aerosols are prepared by the established method of the sol–gel reaction of formaldehyde and resorcinol, as described by Pekala in 1987. Carbon aerogels with mesopores are obtained after supercritical drying followed by pyrolysis of the resorcinol-formaldehyde gel [13]. Presently, different processes for the synthesis of aerogels/xerogels have been developed like drying in the surrounding temperature, drying in a nitrogen atmosphere, microwave drying, or a freeze-drying process for improving the structural properties of the materials to be used for recuperating the performance of supercapacitors. 

Although there are multiple reviews on aerogels [14,15,16,17,18] describing their synthesis and applications [19], their application has not been limited to the fabrication of electrode materials for supercapacitors. Another recent review [20] described the application of porous carbon derived from biomass for the synthesis of electrodes for supercapacitors [20], while we have discussed organic, hybrid, and biomass-derived aerogels and xerogels. There has been another review [21] on graphene-based aerogels in the application of supercapacitors, but we have included xerogels as well as aerogels.

This review encompasses the synthesis and application of aerogels and xerogels within the last 5 years in the field related to supercapacitors. Special emphasis has been placed on bio-derived aerogels addressing the issue of global warming. 

### 1.1. Properties of Supercapacitors

Supercapacitors are categorized as symmetric supercapacitors when similar materials are used for the fabrication of both electrodes, while they are termed asymmetric when the anode and cathode are fabricated with different materials. The capacity to store charge is lower for symmetric supercapacitors because distinct materials favor positive or negative ions present in the solution. By contrast, asymmetric supercapacitors have a high charge storage capacity and a large working potential based on the constituents of different materials in the cathode and anode [14]. Therefore, the storage capacities of both symmetric and asymmetric capacitors can be manipulated by employing different modified aerogels. The function of supercapacitors mainly depends on two significant factors in the electrode composition: the high conductivity of electrons and electrolytes and a larger specific surface area.

#### 1.1.1. Electrical Conductivity

The power density of a supercapacitor is directly influenced by its conductivity, as expressed by Equation (1).
(1)$$P=\frac{{\left(∆E-∆U\right)}^{2}}{4\times ESR\times m_{e}},$$
where *P* = power density per electrodes;

∆E = the window operated by power;

∆U = Ohmic fall;

*ESR* = equivalent series resistance;

$m_{e}$ = active electrode mass. 

The equivalent series resistance (ESR) is affected by several factors such as the resistance of the supercapacitor arrangement, the resistance of the materials influencing the electrodes, and the resistance offered by the electrolyte. Thus, the ESR can decrease if the rate of ion diffusion is high, thereby enhancing the power density in the capacitor [15]. 

The specific capacitance of supercapacitors is given by Equation (2).
(2)$$SC=\frac{It_{d}}{m\Delta V},$$
where *SC* = specific capacitance; 

*I* = charge/discharge current;

$t_{d}$ = time of discharge; 

ΔV = electrochemical potential window;

*m* = active mass.

The capacitance is directly proportional to the surface area of the supercapacitor. Hence, supercapacitor electrodes are typically fabricated using electrochemically inert materials with high specific surface areas. This aids the formation of a double layer with the highest number of electrolyte ions. Graphite, carbon, and metal oxides are materials with high surface activities and are compatible with electrolytes. The relationship between the capacitance of supercapacitors and the surface area is given by Equation (3) [21].
(3)$$\frac{C}{A}=\frac{\varepsilon }{4\pi \delta },$$
where *C* = capacitance of superconductor;

*A* = surface area of electrodes;

*ε* = dielectric constant of electrolyte;

*δ* = distance between the center of the ionic layer to the electrode surface.

The work required to charge a capacitor is related to the energy stored in the capacitor, as expressed in Equation (4).
(4)$$E_{st}=\frac{1}{2}QV,$$
where *E_st_* = energy stored in a capacitor;

*Q* = stored charge; 

*V* = voltage between the negative and positive terminal.

Because the voltage drops during the discharge process, the energy density of the system is given by Equation (5).
(5)$$Energy\;density=\frac{3}{8}C_{eff}V_{r}^{2}\frac{1}{weight\;or\;volume},$$
where *C_eff_* = effective capacitance.

However, this relationship is appropriate only for low power densities and low current values. The stored energy cannot be appropriately analyzed for high-power systems using high power. Thus, the energy storage of supercapacitors was analyzed by performing experiments with constant power over various power densities (generally 100–1000 W/kg). 

#### 1.1.2. Porosity 

The porosity of an electrode is reduced if the inner pores are not interconnected with the surface of the electrode. Therefore, microchannels should be present within the electrodes to connect the inner pores to the surface. The gravimetric capacitance was maximized when the pore size, distribution, and connectivity were optimal. The highest adsorption of ions in the electrode was obtained using a combination of nanosized and microsized pores.

#### 1.1.3. Elastic Properties

The fabrication of thick electrodes enhances the performance of EDLC electrodes owing to their high elasticities. Additionally, the presence of thicker electrodes reduces the use of current collectors and other heavy-cell constituents. Although the volumetric capacitance of thin electrodes is high, the presence of other inactive components in the cell disturbs the overall volumetric capacitance of a cell equipped with thin electrodes. The mechanical stability of the thin electrodes is low during compression, stretching, and bending. Thus, carbon aerogels obtained from glucose with a thickness of 1.5 mm were fabricated as electrodes in EDLC, and they exhibited superelasticity and good mechanical strength [16]. Aerogels with nanotubes were coated with layered graphene, which maintained the morphology of the carbon network and exhibited no deformation, even after many cycles [17]. The layers of graphene coating decreased the electrical conductivity owing to the hoarding of the random layers of graphene at the nodes [18] while enhancing the density of the aerogel compared to that of the uncoated aerogel. Thick carbon aerogel electrodes were examined in the presence of aqueous and ionic liquids as electrolytes, and the capacitance was enhanced with compression. This is because the entire active surface area was employed as the ionic transport distance decreased. The IR drop was subsequently reduced by compression, leading to an increase in the rate of ion transport. The gravimetric capacitance was moderately high, although the thickness of the electrode was based on a high rate of ion transport. The electrochemical performance of the electrodes was stable even after being subjected to numerous cycles of mechanical compression and the release of cells. Graphene aerogels with bubble-like structures [19], honeycomb cellular structures [20], and multiarch structures [21] possess a systematic porous structure with high compressibility and are termed superelastic graphene aerogels. The proper integration of the graphene pore walls contributed to the superelastic properties of the aerogel. The π–π interaction is highly enhanced on the walls of the pores between the different layers of graphene, which provides strength to the walls of the graphene. The value of the elastic modulus was also the highest because the pores were arranged in an ordered manner. A highly elastic electrode material for a supercapacitor was fabricated by embedding polyaniline in superelastic graphene aerogels using an ice template process, followed by the reduction and electrodeposition of polyaniline [22]. The functional groups containing oxygen on the graphene oxide (GO) were reduced, which improved the π–π bonding between the graphene sheets. The aerogels were equipped with an oriented cellular structure with a honeycomb shape in which parallel-stacked graphene sheets formed the walls of the aerogel. The low rate of the freezing propelled the dimension of the cells of the aerogels to be 100 micrometers, thereby facilitating the process of the polyaniline molecules imbibing uniformly on the cell walls. The ordered structure within the aerogel became dense upon compression, retaining its original structure upon release without collapsing. The polyaniline (PANI) remained intact on the surface of the cell walls during compression and expansion. However, a higher amount of polyaniline rendered the aerogel more rigid. The addition of the required amount of PANI enhanced the specific capacitance and imparted good compressibility. These aerogels significantly contributed to the stability, movement of electrons, and conductivity of the electrodes. The electrodes improved the volumetric capacitance owing to their constant gravimetric capacitance and increased density during compression. 

### 1.2. Effect of Different Parameters on Aerogels and Xerogels

#### 1.2.1. Nitrogen Doping

The electrochemical output of supercapacitors can be enhanced by nitrogen doping. Graphene-based carbon aerogels consist of pyridinium and pyridinic nitrogen, which form bonds with potassium ions. A high binding energy indicates that a large number of K ions can be present on the electrode surface, irrespective of the available surface area, thereby improving the capacitance [23]. Pseudocapacitive interactions exist between the functional groups containing nitrogen on the carbon surface and ions of the electrolytes. These interactions led to the development of a high pseudocapacitance [24]. Again, the presence of functional groups containing oxygen also influences the capacitive nature because oxygen supplies the redox activity to improve the capacitance [25]. Additionally, oxygen leads to an increase in the ion adsorption sites owing to ion–dipole attraction, which forms a specific double-layer capacitance boosted by the alteration in the electronic charge density [26]. 

#### 1.2.2. Pyrolysis Temperature 

The pyrolysis temperature used during the synthesis of the aerogel or xerogel controls the morphology of the synthesized material and, hence, affects the properties of the electrodes. The effect of the pyrolysis temperature on the electrochemical characteristics of the aerogel synthesized by the acid-catalyzed polycondensation reaction was studied [27]. Although the surface area of the aerogel was not affected by the annealing temperature, the pore size was significantly influenced by the annealing temperature. The presence of a higher number of mesopores (10–30 nm) was observed when the substance was annealed at 1100 °C. The conductivity increased with the pyrolysis temperature because the resistance between the intergranular boundaries of the aerogels was considerably reduced at high temperatures. However, it is worth mentioning that its conductivity is lower than that of amorphous carbon. The increase in temperature decreased the specific capacitance because the growth of graphene interlayers between the grains was propelled, which obstructed the pores on the surface of the grain. Thus, the pores could not be accessed electrochemically, and surface reduction substantially decreased. The effect of the pyrolysis temperature on the chitosan-based aerogel was studied. A nitrogen-doped graphene aerogel obtained from chitosan contained 3D cylindrical channels before the carbonization process that were disrupted after carbonization [28]. A squashed 3D microstructure was obtained, which was highly porous owing to the transformation of the walls of the channels into flat nanosheets. The 3D porous structure was also retained at the activation temperature; however, the walls were covered with smooth carbon flakes devoid of mesopores and macropores. The walls became rough with an increasing activation temperature and consisted of open pores connected to each other. The morphology of the aerogels was affected by the activation temperature. The thickness of the nanosheet obtained after activation at 700 °C was large with the presence of amorphous carbon with nanopores (2–3 nm). The layer of the walls of the carbon aerogel became thin to form layered graphene at 800 °C and was more elaborately connected. Amorphous carbon particles with diameters (20–30 nm) and nanopores (2–3 nm) were dispensed within the thin and transparent connected layers. These amorphous carbon particles adhered strongly to the transparent layers, giving rise to a hybrid structure of carbon particles and layers. The number of amorphous nanoparticles was decreased at an activation temperature of 900 °C to form graphene with a hexagonal lattice. At 1000 °C, the amorphous carbon particle was eliminated. Thus, the aerogel comprised walls of graphene layers above the activation temperature of 800 °C. The intensity ratio of disordered carbon (I_D_) to ordered carbon (I_G_) was used to analyze the amount of disordered carbon present in the aerogel. This ratio increased with a decreasing activation temperature, indicating that the amount of ordered carbon increased with an increasing activation temperature. Thus, the number of pores and the degree of graphitization in the aerogel increased with an increasing temperature. The aerogel obtained at an activation temperature of 800 °C exhibited an effective performance as an electrode in a supercapacitor based on its adequate, high amount of amorphous carbon content and graphene content. The porous nanostructure imparted a high specific surface area and conductivity. In addition, nitrogen was homogeneously distributed within the aerogel, which improved the device output.

The presence of hierarchical pores (macro-, meso-, and micropores) in gelatin makes it a promising candidate for the synthesis of aerogels for electrodes. Metal ions can impregnate a porous carbon structure for doping, making it an effective electrode material [29]. Therefore, the aerogel was doped with nitrogen at different temperatures (700–900 °C). The porosity and roughness of the aerogels increased with an increase in the activation temperature. The uniform 3D porous structure was disintegrated after carbonization at 900 °C, and the channels were transformed into nanosheets of flat morphology stacked along with a particular orientation. The aerogel formed at an activation temperature of 700 °C formed graphitized carbon, whereas that formed at 900 °C contained a large amount of defective carbon. The interaction between the active sites of carbon and nitrogen enhanced the amount of pyridinic and graphitic nitrogen [30]. The aerogel activated at 900 °C exhibited the lowest energy density and the highest power density. The best electrochemical output was exhibited by the aerogel activated at 800 °C. 

Carbon xerogels were activated with KOH and heated to study their properties [31]. The generation of narrow micropores along with the enhancement of the pore sizes of the existing pores was observed within an activation temperature of 700–1000 °C. Thus, the pore volume and surface area were augmented within this temperature range. The rate of the increase in the melting speed of K_2_CO_3_ and K_2_O, along with the rate of evaporation of K, contributed to the transformation of micropores to mesopores [32]. Chemical activation imparts various defects to xerogels. The intensity ratio also increased for the carbon xerogels with an increase in the activation temperature. The potentials of the electrodes in the organic electrolyte were higher than those in the inorganic electrolyte. This was attributed to the fact that the diffusion rate of the organic electrolytes into the pores of the electrodes was lower because of the larger size of the ions compared to that of the inorganic ions. The compatibility of inorganic ions with the pores in carbon xerogels is much higher [33]. The highest specific capacitance was obtained with KOH (6 M). The xerogel activated at 900 °C exhibited the best output owing to the presence of an adequate number of micropores and mesopores. The adsorption of electrolytes into the micropores is impelled by the mesopores, which augment the accumulation of charge [34]. The specific capacitance of the electrodes increased with longer charging and discharging times.

#### 1.2.3. Freeze Drying Temperature

The aspect ratio of the graphene aerogel could be altered by changing the amount of electrolyte added during electrochemical exfoliation. A graphene sheet with a higher aspect ratio has a large surface area and low thickness. Thus, the pore sizes of the graphene sheets with larger aspect ratios were also larger. Hence, the density of the aerogel can be reduced to increase its conductivity. 

The freeze drying temperature also affected the morphology of the pores in the aerogels. The sizes of the pores of the graphene aerogels ranged from tens of micrometers to submicrometers. The 3D macropores within the aerogels were interconnected in a complex network. The size of the macropores enlarged with a rise in the freezing temperature and reached a size of 60 μm at −20 °C. The freezing rate was accelerated at −200 °C, propelling the rate of nucleation with very little growth of the crystals [35]. Thus, aerogels have small pore sizes because the size of the ice crystals is small [36]. The rate of solidification is very low at −20 °C, imparting less nucleation for the growth of crystals. Thus, fewer large ice crystals were obtained, resulting in the synthesis of aerogels with larger mesopores and macropores (2–50 nm), large specific surface areas, and large volumes. Thus, the freezing temperature plays a vital role in the morphology of the pores in aerogels [37]. The mesopores in aerogels have a cylindrical geometry [38]. The presence of two different types of pores enhances the effective rate of ion transport, thereby increasing the efficacy of the electrodes [39]. Thus the efficiency of the aerogels as electrodes synthesized at −20 °C was higher than that synthesized at −200 °C as the mesopores and the macropores were effectually connected in the aerogels to improve the electrochemical properties [40]. The diffusion of ions was accelerated by the macropores, whereas the mesopores were used for the storage of high energy. The stability of the supercapacitor is also higher. 

#### 1.2.4. pH

Carbon xerogels were prepared from resorcinol and formaldehyde in the presence of NaOH as a catalyst [41]. The output of the xerogels was studied using the response surface methodology (RSM) method. The capacitance and density of the carbon xerogels are dictated by parameters such as the pyrolysis temperature, pH of the solution containing the formaldehyde–resorcinol catalyst, and the ratio of the mass of reactants to the mass of the liquid (R/L). The relationship between the pH and the amount of catalyst is shown in Figure 1.

The gel structure failed to form when the value of R/L was lower than 30%, whereas a pH value lower than 5.5 increased the gelation time and xerogel formation. The capacitance of the electrodes varied even at the same pH. The highest capacitance was obtained with xerogels synthesized at a pH value of 5.7.

## 2. Synthesis of Carbon Aerogels/Xerogels for Capacitors

The unique 3-dimensional (3D) properties of carbon aerogels, such as high conductivity and structured nanonetworks, can be tuned for application in supercapacitors. The three primary steps associated with the synthesis of carbon aerogels are a reaction involving gelation and curing, a drying step to eliminate excess solvent, and the generation of an organic gel, followed by carbonization to fabricate the nanoporous network. However, the traditional method of preparation by the sol–gel process, along with the aging procedure, requires several days to complete. Therefore, various methods have been developed to improve the synthesis process.

### 2.1. Chemical Approach

A highly elastic graphene aerogel was synthesized using an organized chemical approach [42]. A dispersion of aqueous graphene oxide was allowed to accumulate in a reduced graphene oxide (rGO) wet gel by chemical reduction. This was followed by freeze drying to obtain a dry rGO aerogel. The use of iodine and hypophosphorus acid during the chemical reduction process increases the p-p stacking interaction and reduces the electrostatic repulsion between the GO sheets in the wet gel [43]. Again, the rGO aerogel crosslinked with polyvinyl alcohol (PVA) was found to congregate in a different pattern because of the special interaction between the oxygen-containing functional groups of GO and the hydroxyl groups of PVA. Thus, the crosslinked rGO resulted in a higher porosity and a lower density than the simple rGO aerogel, which was unaffected during the freeze-drying and washing processes, as shown in Figure 2. 

The 3D macropores with single-or multilayered pore walls present on the cross-linked rGO aerogel were scattered uniformly, and the restacking process was hindered by the cross-linked interaction of the PVA chains. Although the surface areas of simple and cross-linked rGO were identical, the characteristics of their pores were distinct, with the former possessing mesopores and the latter having macropores. The crosslinked rGO was macroporous because of the PVA chains embedded within the structure, which expanded the volume of the pores, resulting in a rough morphology. The simple rGO aerogel was rigid while the cross-linked rGO was highly flexible due to the cross-linked skeleton, which accommodated the compressive stress and exhibited higher specific capacitance as compared to that of the simple rGO. However, the specific capacitance is lower in organic electrolytes because of the high viscosity and low conductivity of the ions. 

### 2.2. Synthesis on Zn Surface 

Based on the standard values of the reduction potentials of Zn/Zn^2+^ and rGO/GO, the reaction was performed in an acidic medium. Zn is oxidized to Zn^2+^ by releasing electrons that react with GO and reduce it to rGO. The Zn^2+^ surface attracted the negatively charged GO particles and formed a strong contact with the metal. Subsequently, layer-by-layer gelation occurred to form 3D rGO aerogels. The synthesis did not require toxic materials, such as hydrazine, and eliminated the formation of undesired byproducts. The slow growth of the rGO gel on the Zn surface allowed for the possibility of tuning the shape of the aerogel to be tuned depending on the different patterns of the Zn electrode [44]. These reactions are given by Equations (6)–(8).
(6)$${Zn}^{2+}+{2e}^{-}\to Zn$$
(7)$${GO+H}^{+}+e^{-}\to rGO+H_{2}O$$
(8)$${GO+H}^{+}+Zn\to rGO+H_{2}O+{Zn}^{2+}$$

### 2.3. Microwave (MW) Heating

The initial step in the synthesis of carbon xerogels was improved by introducing a unimode microwave heating procedure, which propelled polymerization [45]. The samples obtained after MW irradiation for 40 min exhibited a uniform distribution of particle sizes, which was possible because of appropriate and effective internal heating. This heating accelerates the interaction between the solvents and reagents within the solution by boosting the energy of the reactant molecules. The carbon xerogels are composed of both sp^3^ and sp^2^ hybridized carbon atoms, thereby forming a disordered structure with local crystallization [46]. The 3D hierarchical porous spheres were obtained 100 times faster than the established process and proved to be an effective candidate for ELDC, as they possessed good specific capacitance and cyclic durability without doping and activation. 

Carbon aerogels were prepared from resorcinol formaldehyde using single-mode MW in the presence of water as the solvent and NaOH as the catalyst [47]. The sol–gel reaction was performed at 85 °C and 100 W, followed by pyrolysis at 900 °C for 2 h. The degree of graphitization was low and the carbon consisted of both sp^2^ and sp^3^ hybridized microcrystallites. The spherical nanoparticles formed during the synthesis were found to form channels. These electrodes exhibited electric double-layer capacitance, good electrochemical reversibility, and good coulombic efficiency (CE). The redox reaction of potassium compounds with carbon etched the framework at a high temperature, in which the graphite grains were utilized. Thus, the amorphous structure of the aerogels was enhanced, and a hierarchical structure was developed [48]. The specific surface area of the aerogel was enhanced after activation at 800 °C. 

### 2.4. Electrochemical Exfoliation Method

Graphene aerogel was obtained by electrochemical exfoliation from a suspension containing graphene and used as an electrode. These properties were studied by tuning the exfoliation conditions to manipulate the aspect ratio of graphene. Changes in the concentrations of KOH and H_2_SO_4_ affected the aspect ratios. The size distribution of the pores was controlled by varying the freeze drying temperature. Sulfuric acid and KOH were used as electrolytes during the exfoliation process in the presence of graphene electrodes. A higher concentration of H_2_SO_4_ aids the synthesis of thin graphene sheets. These defects were incorporated because of the highly acidic conditions under which the sulfate anion behaved as an intercalator. Thus, the oxidation of graphene sheets was controlled by adding KOH [49]. The rate of exfoliation was observed by the transformation of the color from transparent to dark. The uniformly dispersed graphene suspension was concentrated to form a gel by interlinking graphene sheets. The graphene sheets were thinner and larger and frozen at different temperatures. The aspect ratio of the graphene sheets was manipulated by increasing their surface area and decreasing the thickness of the graphene sheets. The aerogels with high aspect ratios had good pore structures and low densities. Thinner graphene sheets were obtained at lower KOH concentrations. 

## 3. Aerogels

Single-walled carbon nanotube aerogels were used as electrodes combined with a room-temperature ionic liquid [1-ethyl-3-methylimidazoliumbis (trifluoromethylsulfonyl)imide] as the electrolyte, and their performance was studied [50]. The nanotubes have an open-surface structure with an isotropic network of pores formed by each nanotube. Structural integrity is maintained within the nanostructures [51]. The charge–discharge process was conducted for a shorter time using aerogel electrodes. The electrodes were flexible and exhibited long-term stability. 

Carbon aerogels with hierarchical porous structures were prepared by carbonization, followed by activation using bread as the raw material [52]. These aerogels showed good performance as active materials in a symmetric supercapacitor owing to their good distribution of pores and high specific surface area (1644 m^2^ g^−1^). These parameters were controlled by the activation temperature, and 800 °C was the optimal temperature for activation with the formation of micropores, meso pores, and macro pores. The degree of graphite in the carbon aerogels was increased by KOH activation, thereby enhancing the conductivity of the supercapacitors and diminishing the equivalent series resistance. These aerogels had 95% capacitance retention of over 5000 cycles and showed an energy density of 32 Wh Kg^−1^ with a power density of 400 W Kg^−1^.

In one study [53], an ink with high viscosity, composed of cellulose nanocrystals (CNC) and a silica microsphere suspension, was used in the direct ink writing (DIW) method for the 3D printing of multiscale porous carbon aerogels. The freeze-dried printed lattices were carbonized and activated using KOH, and the silica template was removed to furnish aerogels with varied thicknesses, periodic pore channels, and pores with diameters of 500 nm. The percolation of electrolytes deep into these aerogel-fabricated electrodes was feasible because of their open structures. Pores of 5–50 nm created during freeze-drying owing to ice sublimation boost electrolyte diffusion into the ligaments. The hollow carbon spheres were connected to each other to form carbon flakes in each ligament, as shown in Figure 3. The macropores formed in the hollows served as electrolyte reservoirs, thereby truncating the diffusion path of the ions. This results in the rapid accumulation of charges during the fast charging process. The surface area of the electrode was further improved by the generation of smaller pores via KOH activation. These aerogels showed good capacitance of 71 and 149 Fg^−1^ at scan rates of 200 and 5 mVs^−1^, respectively, in symmetric supercapacitors at a low temperature (−70 °C).

Carbon aerogels doped with nitrogen were produced by an eco-friendly process, in which rhodamine B dye was absorbed by alkaline peroxide mechanical pulp (APMP) fiber aerogels and later carbonized [54]. Although the fibers were stacked owing to the collapse of the pore structures after the absorption of the dye, the 3D network of the synthesized aerogels remained intact. The carbon aerogel exhibited a graphitized geometry with a specific capacitance of 15 g F^−1^ at a current density of 1 g F^−1^. 

Bacterial cellulose (BC) was oxidized using 2,2,6,6-tetramethylpiperidine-1-oxyl (TEMPO) and carbonized using a Zn-1,3,5-benzenetricarboxylic acid (Zn-BTC) soft template to prepare porous carbon aerogels with 3D interconnected nanofibers, excess mesopores and micropores, apparent defects, and large specific surface areas [55]. The electrodes of the symmetrical supercapacitors, fabricated using these carbon aerogels, demonstrated excellent cycling stability with 100% capacitance retention until 65,000 cycles in the presence of a charge/discharge current of 6 Ag^−1^. These electrodes had high power density (0.60 kWKg^−1^) and energy density (15 WhKg^−1^). The specific surface area of the prepared aerogels was substantially increased by the soft template, which was responsible for the broadening of the mesopores and the generation of micropores. Thus, the storage of charge is improved because of rapid ion diffusion through the mesopores and the accumulation of charge through the micropores. 

Graphene-crosslinked polyimide aerogels, formed by mixing polyimide precursors and a graphene oxide suspension, followed by heating in a nitrogen atmosphere, were carbonized to prepare carbon aerogels [56]. The specific surface areas of these carbon aerogels were very high owing to the crosslinking of graphene. The aerogels were covered with pores of different sizes, and the preparation was free from the use of formaldehyde. The equivalent series resistance was small during the rapid charging–discharging process for the prepared carbon aerogels and showed a high capacitance. The π–π stacking between aerogel obtained from polyimide and 3D hierarchical structure contributed towards the low resistance values, as the electrolyte can easily access the surface of the electrode.

### 3.1. Aerogel Beads

A porous graphene bead aerogel was fabricated using a freeze casting method, consisting of channels from the center of the sphere to the surface, with the deposition of a pseudopolymer (polyaniline) into the aerogels [52]. The rGO sheets were layered on both sides using smooth polymers to form a wall sandwiched between the polymers. These beads exhibited good elastic and mechanical properties. The high strength was owing to the increased thickness of the rGO walls and good welding of the walls by polyaniline at the junctions. The thickness of the deposited layers of the polymers could be tailored through the electrodeposition cycles. A thickness of 34 nm was suitable for optimizing the surface area and decreasing the internal resistance. A thicker coating hindered the charge transfer between the electrodes and electrolytes, thereby reducing the capacitance.

### 3.2. Hybrid Carbon Based Aerogels

The properties of the pristine aerogels were improved by synthesizing aerogels combined with different polymers and metal oxides to form hybrid aerogels.

#### 3.2.1. Metal Oxide Hybrid Aerogels

The hybrid V_2_O_5_-rGO aerogels exhibited a higher specific capacitance than the pristine V_2_O_5_ aerogel [53]. Again, the specific capacitance increased with an increasing amount of graphene. Additionally, the specific surface area and conductivity of the hybrid aerogel were enhanced because of the synergistic effect between the aerogels. The V_2_O_5_ aerogel had a rod-like structure, whereas graphene formed semi-transparent thin sheets in the hybrid aerogel. 

Hollow nanoparticles of NiCo_2_O_4_ were used to synthesize a NiCo_2_O_4_@ carbon aerogel by employing a carbon source and sodium alginate as the template [54]. An egg-box-shaped structure was obtained that dictated the diameter of the aerogel. The aerogel has a highly branched 3D structure with different types of pores, which boosts the specific capacitance of the supercapacitor by increasing the diffusion rate along with the movement of electrolytes, ions, and molecules. Additionally, the hollow nanoparticles of NiCo_2_O_4_ were connected to graphitic carbon joints, which improved the electrical contact between the nanoparticles, thereby augmenting the stability during the charging/discharging cycle. Carbon joints induce the hopping of electrons and boost redox reactions [55]. A porous structure is used as the current collector. Despite the good rate of performance, the specific conductance of the aerogels decreased with increasing current density. The increase in the specific surface area of the aerogel enhanced the contact interface between the electrodes and electrolyte ions, which improved the diffusion of ions in the electrolyte, thereby reducing the electron transfer route. Additionally, the hollow nanostructures of the aerogel impart high capacity retention, endowing extra space for accommodating the alteration in volume [56]. 

Another hybrid aerogel was fabricated using rGO and Co_3_O_4_ composites and used as an electrode in capacitors [57]. Pseudocapacitive metal oxide (Co_3_O_4_) nanoparticles were incorporated into the 3D network of rGO aerogel. The capacitance of the composite electrodes was enhanced by the presence of ion diffusion channels in the porous skeleton. Although the electroconductivity of redox-active metal oxides is low, the hybrid aerogel exhibited a high ion diffusion rate, resulting in high conductivity. The negative electrode was composed of rGO, whereas the positive electrode was composed of the composites, which demonstrated good operational stability. 

A composite of a functionalized graphene aerogel and MnO_2_ nanoparticles was synthesized and used as an electrode in supercapacitors [58]. The aerogel was functionalized with p-phenylenediamine (PPD), which transformed the assembly of graphene sheets into a 3D porous structure and aided in maximizing the surface area of the electrode materials. Thus, the performance of the supercapacitors was enhanced by the functionalized materials. 

An asymmetric capacitor was designed using nitrogen-doped rGO (N-rGO) as the cathode and MnO_2_ nanosheets as the anode [59]. The electrodeposited MnO_2_ layers were negatively charged, with water as the intermediate layer and K^+^ as the counterion. The porous morphology of MnO_2_ is appropriate for electrodes because the capillary forces impelled electrolyte mass transfer [60]. Diffusion of the electrolyte was accelerated by the porous and wrinkled morphology of N-rGO at the cathode. GO was reduced with hydrazine using a hydrothermal process to synthesize N-rGO. The oxidation state of the manganese ions was increased from +3.01 to +3.12 V during the charging process and decreased to 0 V during the discharging process. This demonstrates good capacity retention. 

In another study, electrodes were fabricated using a hybrid aerogel composed of MnO_2_ and rGO, which demonstrated high electrochemical performances [61]. The MnO_2_/rGO aerogel was employed as the positive electrode, and a simple rGO aerogel was employed as the cathode. The abundant oxygen atoms on the GO surface aided the adsorption of nano MnO_2_ through H-bonding. The 3D structures consist of thin sheets connected to each other. MnO_2_ nanostructures accumulated within the graphene aerogel, imparting high flexibility. The aerogel demonstrated good capacitive behavior owing to the contribution of pseudocapacity and the double-layer effect. 

A transition metal layer is sandwiched between two chalcogenide layers through van der Waals interactions to form transition metal dichalcogenides (TMD). Thus, the exfoliated TMD sheets were enclosed within a carbon aerogel matrix using sol–gel chemistry [62]. The resorcinol–formaldehyde matrix of the aerogel supported the TMD sheets and enhanced the surface area of the aerogel. Additionally, the aerogel was mechanically stable during the fabrication of the supercapacitor electrodes. The electrodes exhibited good stability during the charge–discharge process, which was attributed to the enhancement of the pseudocapacitance during the cycling process. Another reason for the higher capacitance may be the instigation of additional exfoliation of the TMD [63]. The defective sites in the aerogels tend to charge the trapping sites, thereby aggravating the Fermi level pinning. 

A solid-state symmetric supercapacitor was designed using nanoparticles of Ru_2_O in nitrogen-doped graphene oxide (N-GO) aerogel [64]. The separator and solid-state electrolyte were fabricated using a polymer gel composed of PVA and H_2_SO_4_. Ru_2_O was embedded in the graphene layers while the graphene aerogel remained intact. Increasing the thickness of the electrode enhanced the storage of ions as the surface area increased. However, the storage efficiency was hampered because the thickness created a long diffusion route [65]. Hence, the mass loading was limited to within the threshold. Capacitance retention was 100% after 2000 cycles. The charge on the transparent supercapacitor was stored in Ru_2_O, as shown in Equation (9).
(9)$${RuO}_{x}{\left(OH\right)}_{y}+\delta H^{+}+\delta e^{-}↔RuO_{x-\delta }{\left(OH\right)}_{y+\delta }$$

In another study, supercapacitor electrodes were prepared using an rGO aerogel and ruthenium sulfide (RuS_2_) via hydrothermal synthesis [66]. The rGO flakes were wrinkled and embedded in the spherical nano RuS_2_ structures within the aerogels. The porous structure facilitated electrolyte access. The capacitance of the electrode was increased by RuS_2_ nanoparticles via the Faraday reaction. Annealing increased the crystallinity of the material. The in-plane arrangement of the carbon atoms was enhanced by the reduction performed under hydrothermal conditions, which increased the conductivity. The high stability of the hybrid electrode is attributed to surface activation [67]. 

Another 3D hybrid aerogel was designed using graphene and bismuth tungstate (Bi_2_WO_6_) as electrodes in supercapacitors [68]. The specific surface area of the composite aerogel was larger than that of pristine Bi_2_WO_6_ particles because of its porous and interconnected structure. Additionally, the specific capacitance improved because of the reduced agglomeration of the nanosheets, high specific surface area, and highly conductive network [69]. 

#### 3.2.2. Polymer Hybrid Aerogels

Polyaniline (PANI) was grafted onto the basal planes of an rGO aerogel with the aid of amine groups to synthesize another composite aerogel that was used as an electrode in supercapacitors [70]. Aniline was polymerized in the presence of ammonium persulfate to form PANI, which was grafted onto the aerogel. rGO was diazotized to facilitate the addition of amino groups. Additionally, the diazo groups in GO hindered the aggregation of graphene sheets during reduction. Restacking of the graphene sheets was prevented by placing aryl groups between them. PANI was homogeneously distributed as 3D sheets on the surface of the rGO aerogel. Furthermore, PANI prevented the agglomeration of the graphene sheets and enhanced the surface area. The capacitive output of the electrodes was good, accompanied by good conductivity and accommodation of volumetric alterations. This study proves that diazotization is an efficient process for the synthesis of covalently bonded graphene electrodes. 

Nanoparticles of poly(styrene co-divinylbenzene) were used to synthesize powdered carbon aerogels and as electrodes in supercapacitors with organic electrolytes [71]. The micropores of the powdery aerogels in the electrodes were large (1.3 nm) for easy penetration of the organic electrolytes, which aided the generation of electric double layers. The charging–discharging properties were enhanced by the nano-network of macropores and mesopores within the range of 10–100 nm, which also facilitated the movement of ions. The powdered form of the aerogel enhanced the binding properties of the conductors and the binding of the polymer with the current collector. The electrodes exhibited a capacity retention ratio of 94% over 300 cycles and good cyclic stability. The rigid hypercrosslinking of the powdery polymer aerogel was well retained even after carbonization.

Triazine-based conjugated microporous polymers were mixed with graphene aerogels to synthesize hybrid aerogels for the fabrication of electrodes and to increase their conductivity [72]. The layered structure of the aerogel was highly cross-linked and surrounded by a porous conjugated microporous polymer. The edges of the aerogel or polymer layers contained pyrrolic or pyridinic nitrogen. The amount of pyridinic nitrogen increased, whereas that of quaternary nitrogen decreased upon treating the electrode with ammonia at high temperatures. This improved the supercapacitative output. There was no voltage drop, and good conductivity was obtained. Annealing in an ammonia atmosphere enhances the performance. The mesopores and micropores of the conjugated polymer reduced the diffusion route between the surface of the electrodes and the external electrolyte. Graphene behaves as a conductive framework. The charge transfer resistance was lower at the interface of the electrode and electrolyte, and faster ion diffusion was observed owing to the higher ratio of nitrogen [73]. The cyclic stability and energy density were also superior. 

Composite monolith electrodes were fabricated for supercapacitors using phenolic resins and polyacrylonitrile fibers [74]. The presence of intraparticle pores with sizes less than 5 nm, along with the existence of interparticle pores formed by specially packed carbon particles, imparted good electrochemical properties. The surface area and capacity were positively influenced by intraparticle pores, whereas the conductivity was enhanced by interparticle pores. The surface area was enlarged with a carbonization temperature below 700 °C, while it was decreased with a rise in the temperature as the channels collapsed and became irregular. Conversely, the resistivity increased with an increase in the carbonization temperature.

### 3.3. Aerogels from Green Biomass 

#### 3.3.1. Cellulose-Based Aerogels 

Carbon aerogels comprise carbon nanotubes and graphene and have been extensively employed as electrodes in supercapacitors owing to their good intrinsic electrochemical properties [75]. Although graphene-based supercapacitors are extensively used, they face certain drawbacks such as the restacking of graphene nanosheets, which ultimately hinders the diffusion of electrolyte ions [76]. This limitation was overcome using cellulose nanofibril (CNF)-based aerogels as electrodes in capacitors. Additionally, hierarchical composites of porous carbon obtained from carboxymethyl cellulose, polysaccharides, citric acid, and bacterial cellulose have shown excellent performance as supercapacitor materials. The high porosity of CNF aerogels, along with their good electrolyte absorption properties and high specific surface areas, make them exceptional materials with good electrochemical properties [77]. The surface contact between the electrolytes and electrodes was significantly enhanced owing to the hydrophilicity of the CNF aerogels, which also generated channels for the diffusion of electrolyte ions [78]. Flexible electrodes were fabricated using CNF/reduced graphene oxide (rGO)/carbon nanotubes (CNT) hybrid aerogels, as shown in Figure 4 [79].

GO behaves as a surfactant for the dispersal of CNT [80]. GO nanosheets are hydrophilic, constituting a large number of oxygen atoms, thus enabling the formation of a homogeneous solution with CNF [81]. A channel was produced between the multiple layers of graphene nanosheets by the CNT, which aided the movement of ions [82]. Toxic reagents such as hydrazine were avoided during the synthesis. The densities of the samples decreased because of the elimination of oxygen-containing functional groups while retaining the 3D porous structure of the aerogels. The aerosol films were prepared without electroactive additives, binders, or current collectors. The efficient output of the capacitor was also because the aggregation of graphene was hindered by both the CNT and CNF. Again, the diffusion of ions was accelerated by the highly porous structure of the electrodes, as CNF behaved as the nanoreservoir of the electrolytes, generating diffusion channels. 

Mechanically flexible carbon-fiber aerogels with high surface areas were prepared from natural cotton via carbonization and KOH activation, as shown in Figure 5 [83]. The movement of ions and electrons was facilitated along the tubular and helical carbon fibers, forming a 1D skeleton. Moreover, KOH activation generated numerous nanopores on the surface of the fiber walls for charge storage. A 79% of capacitance retention was observed at a current density of 100 A g^−1^. The electrodes fabricated with the prepared carbon fiber aerogel in a symmetric supercapacitor exhibited a small relaxation time constant of 0.56 s. The electrodes retained their electronic and ionic transport characteristics during the long-term cycling. Thus, different pseudoactive electrodes can be fabricated using these carbon fiber aerogels based on their compression recovery properties. 

Carbon nanofiber aerogels were synthesized from wood nanofibril cellulose in the presence of p-toluenesulfonic acid as a catalyst during pyrolysis [83]. The aerogels were used as electrodes in supercapacitors in the absence of binders. Ultrafine carbon nanofibers that remained interconnected and tangled after pyrolysis were formed. The uniform structure of the cellulose nanofibers was preserved even after strong pyrolysis, and agglomeration into sheets and spheres was hindered [84]. The volume of the aerogel shrank after pyrolysis at 800 °C with 26.3 wt%. of carbon residues. Electrical conductivity and the degree of graphitization increased with an increase in the pyrolysis temperature. The aerogels had a high compressive strength and were mechanically stable. The TBA freeze-dried procedure formed a network of 3D nanofibers with high surface area after pyrolysis at 800 °C. These aerogels exhibited an electric double-layer capacitive nature and good cyclic stability. Carbon nanocellulose fibers obtained from wood were combined with mixed-valence MnO_x_ to form composite aerogels that were used as bioelectrodes in supercapacitors [85]. The numerous hydroxy groups on the fibrous nanocellulose, along with the high aspect ratio, initiated the formation of the gel network owing to the tangling of the fiber network [86]. The suspension of the nanocellulose was mixed with Mn(OAc)_2_ and freeze-dried for the generation of the aerogel followed by calcination at 850 °C (Figure 6). 

The existence of both acceptor and donor sites in the microstructures, along with defects, reveals a high charge storage capability. Redox reactions, along with ion intercalation and deintercalation, occur on the surface of MnO_x_ during the charge–discharge and charging processes. The electrical characteristics were enhanced by increasing the MnO_x_ mass below 70%. The charge–transfer impedance was diminished by the channels obtained from the intricate web-like structures. 

Bacterial cellulose (BC) was pyrolyzed in microwave plasma for 15 min to fabricate an aerogel with crosslinked carbon nanofibers [87]. The carbonized bacterial cellulose aerogel was used as an electrode in electrochemical capacitors, which exhibited better electrochemical characteristics. The main advantages of BC over natural cellulose are that it is pure, it possesses a higher degree of crystallinity, and the size of the fibers ranges from 10 to 50 nm. The plant-derived cellulose nanofibers (CNF) do not remain interconnected. However, CNFs derived from BC are highly cross-linked with a large surface area and exhibit good conductivity within the electrode after pyrolysis [88]. The distributed pores were macro- and meso-sized, which allowed for good movement of ions through the electrode mesh, possessing stumpy ionic resistance. Dead-end pores, micropores, and slit pores were absent, which enhanced the ionic conductivity. The fibers in the aerogels are highly interconnected, which is extremely effective for a strong response to the frequency of the 3D disseminated electrodes. The CNF possessed either a sheath or tubular structure. The thickness of the electrodes influences the frequency response and capacitance of the device. The volumetric capacitance was constant for both thin electrodes, whereas the areal capacitance increased. A thick electrode exhibited a higher areal capacitance but a lower volumetric capacitance than thinner electrodes. This study revealed that the constituents of the plasma gases used during pyrolysis affected the frequency response. The etching was highly enhanced by the pure hydrogen in the plasma, thereby inducing the formation of micropores. A hybrid aerogel was synthesized by integrating NiMoO_4_@Co_3_O_4_ with a carbon aerogel obtained from cellulose and was used in an asymmetric supercapacitor. The nanorods formed from NiMoO_4_ were filled within a 3D carbon aerogel by a hydrothermal method and were finally loaded with polyhedral nanocrystals of Co_3_O_4_ [89]. The NiMoO_4_ nanorods surrounded each aerogel, whereas ZIF-67 was deposited within the voids. The dodecahedral structure was maintained after pyrolysis, and Co_3_O_4_ formed a porous nest-like structure. The composite aerogels exhibited three different network structures. The interconnected nanofibers of the aerogel formed small bundles of filaments to form the primary network, which acted as a backbone and diffusion channel for the ions in the electrolytes. The NiMoO_4_ nanorods formed the second network, which connected the ternary hierarchical structure, thereby contributing to a large surface area for the loading of the active materials. This also enhances the pseudocapacitance properties. The final network consisted of nanocavities and channels formed by the nanocrystals Co_3_O_4_ which possessed numerous active sites for charge storage. Thus, redox reactions are boosted by the accelerated rate of electron transport between the electrolyte and electrodes [90]. The diffusion routes were diminished owing to the large specific surface area of the Faradaic reactions, which improved the supercapacitor performance. There is a possibility of multiple redox reactions owing to the co-presence of Ni and Co. Thus, the composite is effective as a pseudocapacitor material. 

Nanocellulose was derived from *Phyllostachys pubescens* to synthesize graphene nanocellulose aerogels using a hyrothermal metho [91]. This composite electrode material exhibited excellent porosity, which facilitated the entry of the electrolyte into the electrodes, thereby improving electrochemical performance. Additionally, the cyclic performance of the electrode was also better, as there was no agglomeration of graphene, with a uniform distribution of pore sizes as the hydrophilic cellulose nanofiber formed a 3D skeleton. 

A composite aerogel was prepared using cellulose nanofibrils, polyaniline, and graphene nanosheets as the electrode materials. The electrodes exhibited good electrochemical performance owing to their high porosity and conductivity [92]. The in situ polymerization process generated high-quality aerogels compared with those generated by the mechanical mixing method. The cattail cellulose microfibers were pretreated with sodium chlorite followed by pyrolysis at 900 °C in a nitrogen atmosphere [93]. Pyrrole was polymerized in situ and adhered to the surface of the synthesized aerogels (Figure 7). This composite aerogel was used as an electrode devoid of any binder, thereby reducing the resistance of the interface and enhancing the capacitance. The polypyrrole particles induced the formation of a rough surface owing to the agglomeration of particles over the carbon fibers and formed channels for improved conduction. 

Another report [88] described the synthesis of multilayered nanosheets (thickness = 10 nm) by the carbonization of dicyandiamide and glucose, which were dispersed in a suspension of cellulose nanofibers. This suspension was free-cast, freeze-dried, and carbonized to obtain nitrogen-doped carbon aerogels with a cross-linked network, an elastic structure, and a stable structure with wavy layers. These multiple continuous and parallel layers are capable of resisting geometric deformation under compression and effectively transferring stress. These flexible biomass-derived aerogels have the potential for use as supercapacitor materials. 

In another study, cellulose nanofibrils were oxidized using 2,2,6,6-tetramethylpiperidinyl-1-oxyl (TEMPO) and carbonized to form carbon aerogels [91]. The synthesized carbon aerogels were doped with nitrogen by reacting with urea at a high temperature, followed by a hydrothermal reaction to embed manganese oxides in the aerogels. The prepared composite aerogels exhibited good pseudocapacitance because of the presence of manganese oxide and nitrogen. The electrolytes diffused easily because of the presence of abundant pores and large specific surface areas. Pyridine and pyrrole nitrogen increased the specific capacitance of the aerogels, and the conductivity was boosted by graphite nitrogen. These fabricated aerogels were used as electrodes in an asymmetric supercapacitor (LiPF_6_ as electrolyte, activated carbon as the other electrode) which showed a power density and energy density of 600 W kg^−1^ and 23 W H kg^−1^, respectively, at 0.5 A g^−1^. Moreover, capacitance retention was 80% after 1000 cycles at 10 A g^−1^.

#### 3.3.2. Fruit Biowaste

Aerogels fabricated from durian and jackfruit scraps were used in the ELDC setup and demonstrated excellent performance [94]. The synthesis of bioaerogels is shown in Figure 8.

The bulk material of aerogel obtained from durian consisted of macroscopic channels along with the presence of macropores with diameters of 5–15 μm [95]. Trench-like structures and small pores were observed along the length of the channels, which were 6 and 800 nm apart. Conversely, the aerogel obtained from jackfruit revealed equal diameters (20 μm) of the pores, which were connected through gaps. The aerogels obtained from both fruits were amorphous with graphitized areas. The capacitances of the electrodes were high because of their large surface areas and highly porous morphologies. Some nitrogen-containing functional groups and chlorine were present in the aerogel. The specific capacitance was enhanced in the presence of negatively charged pyridinic nitrogen at the defect sites. This phenomenon occurred because of the interactions of the pseudocapacitive characteristics, in which the ions could easily interact with pyrrolic nitrogen. Additionally, the conductivity was enhanced by positively charged graphitic nitrogen [96]. Chlorine was strongly bound to the aerogel surface and could not be easily removed. The nitrogen content of the durian fruit aerogel was higher than that of the jackfruit aerogel. The presence of elements such as oxygen, nitrogen, and chlorine on the aerogel surface boosted the affinity of the surface towards the electrolytes and affected the pseudocapacitance, thereby increasing the net capacitance [97]. The durian-derived aerogel demonstrated a good output owing to its large surface area and mesoporous morphology compared to the jackfruit aerogel. Additionally, the positive charge density of carbon was enhanced by the presence of nitrogen, which in turn increased the wettability and polarity of the surface [98]. The cyclic stabilities of both aerogels were significantly improved. 

In another study, pears were treated hydrothermally and later carbonized to produce lightweight graphene-based aerogels with high surface areas [99]. The highly porous aerogels are good for storing energy, whereas the interconnected 3D network aids the movement of electrons and ions. The specific surface area of the prepared aerogels increased to 2300 from 1000 m^2^ g^−1^ through KOH activation, resulting in enhanced cyclic stability, energy density, and elevated specific capacitance. This improvement was attributed to the fact that the amorphous carbon lying deep within the graphene layers was removed, and the amount of graphene in the activated graphene aerogels was enriched. These graphene aerogels, examined in a coin-cell-type supercapacitor, demonstrated an energy density of 57 Wh kg^−1^ and a power density of 620 kW kg^−1^ with 10,000 charge/discharge cycles at a current density of 2 A g^−1^ and 83% capacitance retention.

Three-dimensional carbon aerogels prepared from radishes as a precursor through hydrothermal treatment, freeze drying, and carbonization were doped with MnO_x_ nanoparticles [100]. A mosaic space was available in the formed carbon aerogels to embed the MnO_x_ nanoparticles. The amount of MnO_x_ nanoparticles improved the degree of graphitization, thereby increasing the gravimetric specific capacitance. Excess micropores were produced by increasing the number of MnO_x_ nanoparticles. A three-electrode supercapacitor (KOH as electrolyte) fabricated with these aerogels showed an energy density of 248 Wh kg^−1^, a power density of 4780 W kg^−1^, and a high gravimetric specific capacitance of 557 F g^−1^ at a current density of 1 A g^−1^. Therefore, these aerogels have the potential to be used as inexpensive electrode materials. 

Two different aerogels were prepared from carrot and pumpkin by combining them with palmitic acid and thiolene resin to form stable phase change materials by a hydrothermal process, followed by a sintering process [101]. These phase change materials are important for thermal energy storage systems because of their high-density energy storage [102]. The thermal conductivity of the palmitic acid/thiolene resin composite was highly enhanced by the aerogels obtained from pumpkin and carrot. However, the thermal conductivity of the composite was at its maximum for aerogels obtained from carrot by sintering at 1000 °C. The composite prevented leakage owing to the porous structure of the aerogels.

#### 3.3.3. Lignin-Based Aerogel

Lignins derived from organosolv and kraft were used to synthesize lignin-based carbon aerogels by the polycondensation of lignin and formaldehyde using sol–gel methods [99]. Among the two lignins, organosolv lignin reacted well with formaldehyde because of its low polydispersity and molecular weight. Moreover, the organosolv required a gelation time of 2 h compared with that of kraft lignin, which required a considerably longer gelation time (16 h). However, formaldehyde attacks the meta-positions of the aromatic rings in lignin, which drastically affects the gelation procedure. Organosolv lignin is mainly composed of sinapyl alcohol, whereas kraft is composed of p-coumaryl alcohol and coniferyl alcohol. Thus, the aerogel obtained from kraft is improved, indicating that the structural features of lignin are more important than other parameters, such as molecular weight and polydispersity. Both mesopores and micropores are present in the aerogel obtained from kraft, whereas only macropores are present in the organosolv-derived aerogel. Thus, the specific surface area can be increased by increasing the number of reaction sites. These properties can be exploited using aerogels as electrodes in supercapacitors.

Ice templating and carbonization were performed for the synthesis of kraft lignin-based aerogels using kraft lignin and cellulose nanofibers oxidized by TEMPO (TOCNF) [100]. The transfer of energy or mass was enhanced by the formation of an anisotropic longitudinal structure during ice templating [101]. The horizontal expansion of the ice crystals was inhibited by the increased viscosity of the suspension formed by lignin and TOCNF. Thus, an increase in the concentration of TOCNF reduced the size of the macropores [102]. The numbers of mesopores and micropores increased with an increase in the amount of TOCNF, along with the surface area. The anisotropic porous network was unaffected during the carbonization process, but the cell walls shrunk and wrinkled with a reduction in the amount of oxygen but with an increase in the carbon content. The transfer of Na from the inner walls to the surface has also been observed after carbonization [103]. The transfer of ions and electrons is high, leading to a high capacitive ability based on the large surface area and porous network [104]. Conversely, the hydrophobic electrodes exhibits high resistance to charge transfer. 

One report described the synthesis of carbon aerogels from cellulose fibers (separated mechanically) using either soda or kraft lignin [105]. These cellulose nanofibers possess large diameters and, hence, impart suitably sized pores to the prepared aerogels as sacrificial templates during carbonization. Parameters such as pH and viscosity are important for suspensions containing lignin and cellulose nanofibers during ice templating. The morphology and growth velocity of the ice crystals are dictated by these parameters, which, in turn, affect the structure of the pores in the prepared aerogels. Both lignin-based aerogels exhibited 3D macropores in the form of axially placed channels. These macropores decreased the distance between the diffusion paths of the electrolytes and the active sites. Thus, the resistance to ion transport decreased, enabling the creation of effective electrical double layers [106]. The viscosity of the suspension was decreased by reducing the number of cellulose nanofibers, which in turn dilated the size of the macropores. The number of nucleation sites was increased by increasing the number of cellulose nanofibers in the suspension, resulting in the creation of smaller-sized macropores. Additionally, the electrochemical properties of both the lignin-based aerogels were examined as materials for a two-electrode supercapacitor, revealing that the kraft linin aerogel was more effective than the soda lignin aerogel. An equal distribution of macropores and micropores in the aerogel obtained from 60 wt% kraft lignin imparted the best electrochemical characteristics owing to the hierarchical morphology of the pores. The increased size of the macropores enhanced the contact resistance, thereby increasing the charge-transfer resistance. The main objective of this study was to determine the morphology of the prepared aerogels, and whether their electrochemical properties could be modulated by optimizing the amount of cellulose nanofibers. 

Carbon aerogels obtained from lignocellulosic materials were prepared by templating on ice, freeze drying, and carbonization. Graphene dots were embedded into the prepared aerogels and examined as supercapacitor electrodes [107]. The synergistic effect of the pseudocapacitance and enhanced surface area in the aerogels modified with graphene dots improved the electrochemical performance. This was due to the introduction of excess functional edges and groups. The properties of both pseudocapacitance and electrical double-layer capacitance were demonstrated using these aerogels. Additionally, their equivalent series resistance was higher than that of the pristine aerogels because the contact resistance between the current collector and the electrode was higher. The high rate of ion diffusion within the electrodes was boosted by the graphene dot-embedded aerogels, thereby imparting pure capacitive characteristics with a specific capacitance of 180 F g^−1^. The energy densities were 3.8 and 6 Wh kg^−1^ at lower densities of 500 and 50 W kg^−1^, respectively.

### 3.4. Coal

In another study [108], graphene oxide was derived from bituminous coal using a modified Hummers’ method. The characteristics of graphene oxide derived from coal differ from those of GO obtained from natural graphite. Coal-derived graphene oxide possesses many structural defects with strongly bonded graphene layers and a large surface area, making it highly porous. A solution of this coal-derived graphene oxide was combined with carboxymethyl cellulose by a hydrothermal reaction, followed by freeze drying to synthesize graphene oxide aerogels to eliminate dyes. The prepared aerogels exhibited high mechanical stabilities, good absorption capabilities, and defects in the graphene framework. In another report [109], titanium oxide/bituminous coal-based graphene aerogels were synthesized using ethylenediamine as a crosslinker and carboxymethyl cellulose as a filler via a hydrothermal reaction and freeze drying techniques. These hybrid aerogels exhibited exceptionally low density, abundant oxygen-containing functional groups, good mechanical strength, and a highly porous structure. They have been used for the effective removal of dyes from wastewater. Coal-derived graphene aerogels were prepared using a freeze-drying process and used as an absorption material and an interfacial solar vapor evaporator [110]. The graphene aerogels are highly porous, fire-resistant, compressible, and elastic. In addition, the aerogels exhibit strong absorption capabilities.

Three-dimensional hierarchical carbon aerogels have been synthesized by the carbonization of hydrogels obtained from freeze-dried PVA and coal [105]. The mesopores and micropores in the aerogel were modulated by altering the PVA-to-coal oxide mass ratio. The pore structure of the aerogel was influenced by the amount of PVA because the stacking of the crosslinks was boosted by the presence of an excess amount of PVA. The highest graphitization degree was found for the aerogel formed at 800 °C which imparted maximum conductivity. The pores disintegrated at higher temperatures owing to the decomposition of PVA. Faradic pseudocapacitance was imparted in the electrolytes owing to the presence of pyrrolic-N and pyridinic-N, thereby improving the capacitance [106].

## 4. Carbon Xerogels

Porous xerogels equipped with functional groups like lactone and anhydride on the surface were designed employing a suspension of graphene oxide in water. The porous xerogels equipped with functional groups like lactone and anhydride on the surface were designed employing a suspension of graphene oxide (GO) in water. Thus, the amount of oxygen-containing surface groups is much less, and the average surface area of the xerogels was 1600 m^2^ g^−1^ with the presence of micropores. The sizes of the mesopores were dictated by the amount of GO added during the gelation step under MW irradiation. The collapse of the porous structure under MW was prevented by the addition of a limited value of GO (0.11 wt%). The mesopores experiencing small shrinkage during the process of carbonization and activation [107] were hindered due to the addition of GO. The random dense packing of the carbon nanosheets within the xerogels was devoid of gaps which enhanced the conductivity as it achieved the threshold of the electrical percolation fast. The main property required for a capacitor to possess high conductivity is that the ions should be capable of moving freely through the macropores with accessible diameters and reach the micropores where the charges are accumulated [108]. The carbon xerogel was doped with cobalt and was used as electrodes which exhibited better performances due to large pores formed by the movement of the Co through the xerogel [109]. 

### 4.1. Metal Organic Xerogels

High-surface-area metal organic xerogels (MOXs) were synthesized from MOF by solvothermal synthesis [110]. The nanoparticles of the MOF aggregated to form the xerogels due to van der Waals interaction, π–π interactions, and hydrogen bonding (Figure 9) [111]. The positive electrode was fabricated of MIL-100 (Fe), while the negative electrode was fabricated of MIL-100 (Al). The assets of the MOXs were low densities, high surface area, large channels, and porosities which could be manipulated. The carbonization of the xerogels for the anode was carried out at 480 °C and above 570 °C to form partially decomposed and completely decomposed carbon frameworks, respectively. The partially decomposed structure of xerogels consisted of large particles enclosed within a thick shell of carbon. The oxidation of the outer shell takes place due to the reduction of Fe_3_O_4_ to Fe by carbon, which reduces the conductivity through the thick carbon shell [112]. At higher temperatures, the nanoparticles were scattered homogeneously over the carbon. However, at very high temperatures, the surface of the Fe was converted to iron carbide which decreased the capacity of string charges based on its poor redox activity. The MOX activated at 700 °C was found to be effective due to the formation of a thin layer which prevented any change in the structure of the nanoparticles on the electrodes. Additionally, the morphology was polycrystalline with a mixture of Fe and Fe_3_O_4_. The iron was present in Fe^2+^ and Fe^3+^ state which boosted the conductivity. The Fe was oxidized to Fe_3_O_4_ in contact with the electrolyte which improved the performance. The high capacitive properties were attributed to the participation of the redox active particle present within the hierarchical matrix of carbon. The charge storage of the double layer was contributed by the carbon which further heightens the efficacy with capacity retention during the cycles of the charge–discharge procedure. The MOX-Al was carbonized at 1000 °C to obtain a high surface area and further treat it with NaOH. The diffusion of the ions was accelerated due to the disordered structure of graphitic carbon. The micropores and mesopores are interconnected to demonstrate high capacitance. The superconductor exhibited good cyclic stability. 

### 4.2. Bio-Xerogels

Bio-xerogels were synthesized using lignin from sugarcane bagasse through sol–gel polymerization [113]. The pore sizes in the xerogels were limited to 4 nm which were distributed heterogeneously. The cross-linking and the kinetics of the polymerization was dictated by the pH of the dissolved solution. Thus, the parameters like particle size, porosity, and morphology were influenced by the dissolution pH [53]. The addition reaction was predominant at a high pH instead of the polymerization condensation reaction, leading to structures with low-quality pores [114]. The lignin was crosslinked with formaldehyde and resorcinol during the polymerization condensation reaction by forming random C-C bonds and ether linkages [115]. The large size of the lignin influenced the formation of macroporous xerogels, as good packing was prevented due to the large size which also hindered shrinkage [116]. The bio-xerogels were activated by H_3_PO_4_, where the enhancement of the lignin increased the number of acid groups. An increase in the volume of the micropores enhanced the gravimetric capacitance. The mesopores present in the xerogels exhibited low ESR and better retention capacity at the highest value of current. 

## 5. Comparison of the Efficacy of Graphene Aerogel or Xerogel Electrode

A comparative study was conducted on the performance of the close-packed xerogel and open-porous graphene aerogel [117]. Both the aerogel and the xerogel were synthesized from graphene suspension. The ordinary graphite was exfoliated electrochemically to form the suspension of graphene in the presence of KOH (30%) and H_2_SO_4_ (0.2 M). The highly concentrated graphene sheets were obtained in the absence of any surfactants (Figure 10). The graphene aerogel exhibited better performance than that of xerogel as electrodes in supercapacitors as the aerogel possesses a high surface area along with mesopores and macropores. The dispersed solution was concentrated whereby the interlinking graphene sheets formed the graphene gel. The solvent in the gel was evaporated to form the xerogel, while the gel was frozen, and the frozen solvent was sublimed to maintain the porous structure and form the aerogel [118]. The low values of ID/IG indicated that both the gels were less defective and were of high quality with a faster rate of diffusion of ions. The pore sizes of aerogel were smaller with a 3D interconnected macroporous structure than those of the xerogel, which consisted of a close-packed structure. The morphologies of the resultant structure depended on the process of drying. The aerogels were designed as electrodes devoid of conductive agents and binders. The thickness and density of the electrode films for xerogels were 2 μm and 400 mg cm^−3^, while those of aerogels were 80 μm and 10 mg cm^−3^, respectively.

The capacitance of the aerogel supercapacitor was found to be 3.6 times higher than that of the xerogel supercapacitor. The xerogel tends to agglomerate while drying, which poses a limitation in the process. However, the electrode resistance of both the xerogel and aerogel was similar. The density of the aerogel is 40 times lower than that of the xerogel. The capacitance due to diffusion was dominated by the xerogel supercapacitors. In contrast, the transport of ions was faster in aerogel electrodes as the charge moved at a faster rate into the storage sites of the electrode, thus enhancing the capacitance. Additionally, the lifetime of the supercapacitors with aerogel was higher due to their cyclic stability and higher coulombic efficiency. A comparison table of the properties of the aerogel and xerogel is shown in Table 1.

## 6. Conclusions and Future Perspectives

This review highlights aerogels and xerogels as promising candidates for the synthesis of highly efficient supercapacitor electrodes. Various synthesis methods have been discussed, among which microwave heating is a promising technology owing to its short processing time. Additionally, uniform heating enhances the quality of aerogels and xerogels. The pyrolysis temperature significantly influences the morphology of the gels, suggesting substantial potential for improving the morphology of the electrode material through precise tuning of the pyrolysis temperature.

Although pure aerogels and xerogels exhibit noteworthy electrochemical properties, their performance can be further enhanced by synthesizing hybrid aerogels and xerogels incorporating different metal oxides, polymers, and ionic liquids. There is ample opportunity to discover other hybrid gels with improved performances on a commercial scale.

Flexibility is a crucial requirement for future energy storage devices. The capacitive performance should not be compromised by electrode bending. Thick electrodes with good mechanical and elastic properties, such as single-walled carbon nanotube aerogel electrodes, have demonstrated effective performance in the presence of ionic liquid electrolytes. Aerogels derived from plant biomass also exhibit favorable elastic properties, highlighting the need for research on novel materials with electrochemical stability under stress to advance the development of flexible supercapacitors. These flexible electrodes are promising candidates for use in various electronic devices and electric vehicles.

Chitosan-derived aerogels have a hierarchical porous structure that is advantageous for supercapacitor electrodes. This suggests a promising avenue for future investigations on different biomass materials for conversion into aerogels or xerogels with hierarchical porous structures for electrode applications. Excellent charge–discharge properties are crucial for proper electrode functioning in supercapacitors, and both pristine and hybrid aerogels and xerogels exhibit excellent performance in this regard.

Despite the inherent potential of aerogels and xerogels as electrode materials in supercapacitors, their synthesis requires further improvement for practical applications. It is essential to scale up the synthesis from laboratory-scale operations for industrial applications. The use of biomass as a precursor could potentially decrease costs compared to chemical precursors. The commercial production of these materials holds promise for technological advancements in energy applications and storage.

The higher surface area of these gels compared with that of bulk materials makes them more potent storage materials, warranting industrial exploitation for the fabrication of high energy storage devices. Further development of aerogels and xerogels from organic waste for designing next-generation capacitors could also serve as a meaningful initiative for recycling substantial waste materials generated globally.

## Figures and Tables

**Figure 1 polymers-16-02848-f001:**
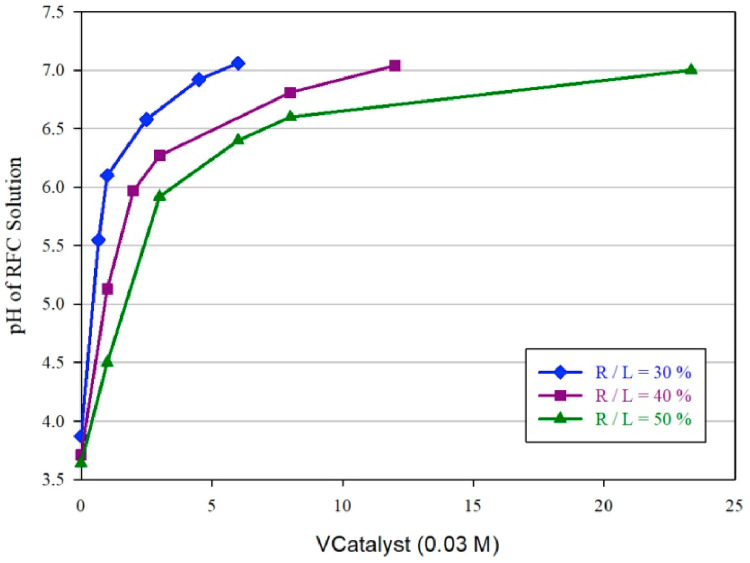
Relation between the pH and the amount of catalyst [41].

**Figure 2 polymers-16-02848-f002:**
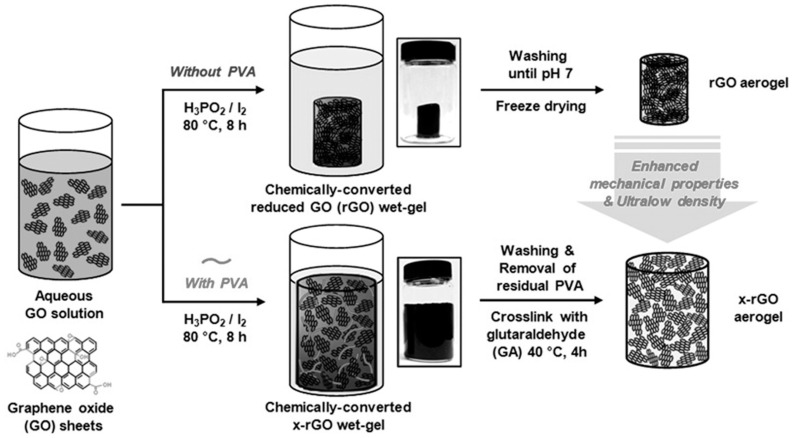
Schematic of the synthesis of the aerogels [42].

**Figure 3 polymers-16-02848-f003:**
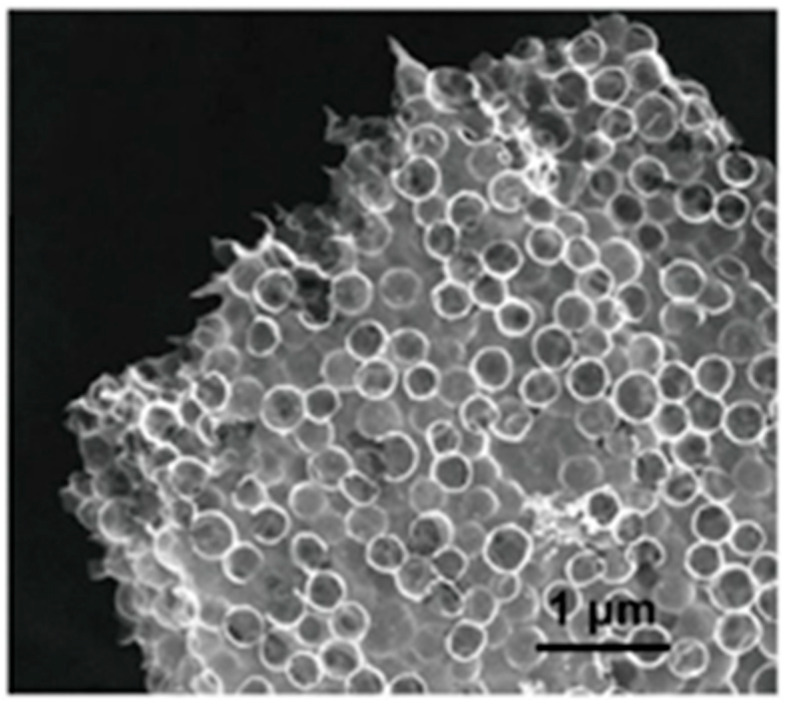
SEM image of the carbon aerogel.

**Figure 4 polymers-16-02848-f004:**
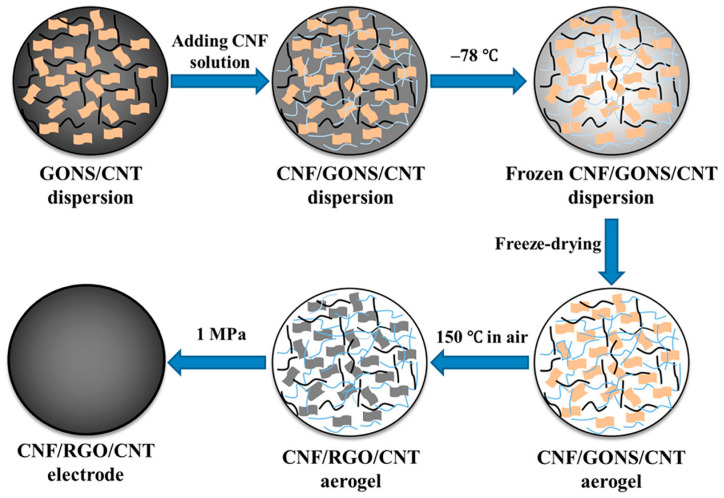
Schematic of the fabrication process of CNF aerosol/rGO/CNT electrodes [79].

**Figure 5 polymers-16-02848-f005:**
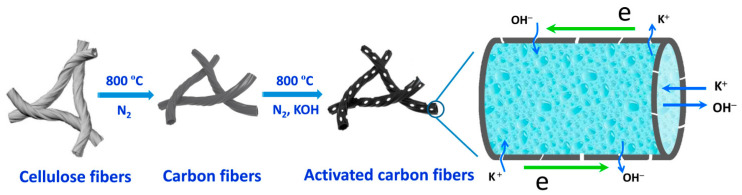
Synthesis of carbon fiber aerogel from natural cotton.

**Figure 6 polymers-16-02848-f006:**
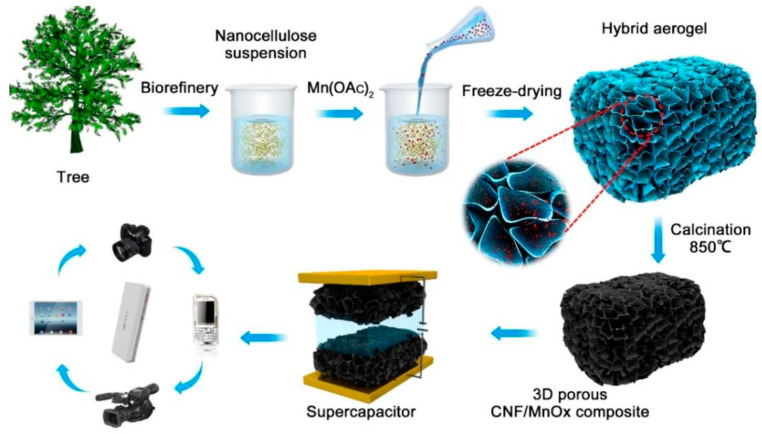
Schematic representation of the preparation of aerogels from wood [85].

**Figure 7 polymers-16-02848-f007:**
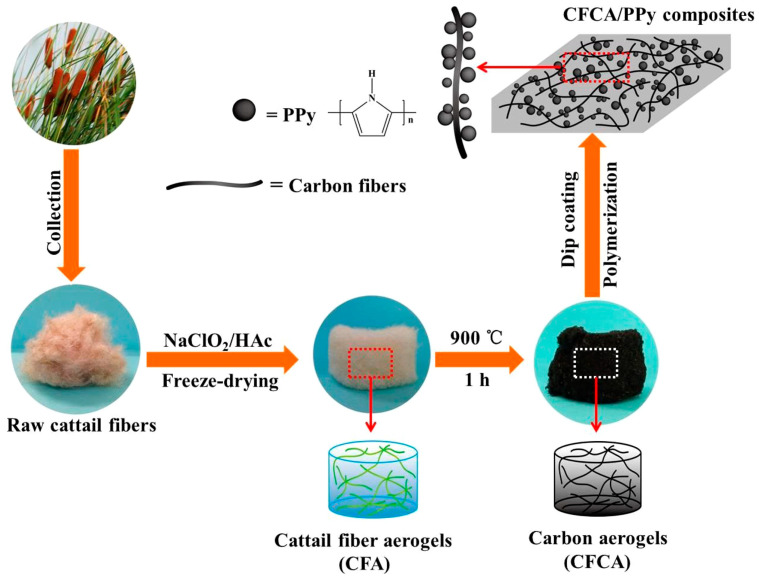
Schematic representation of the synthesis of cattail aerogels [93].

**Figure 8 polymers-16-02848-f008:**
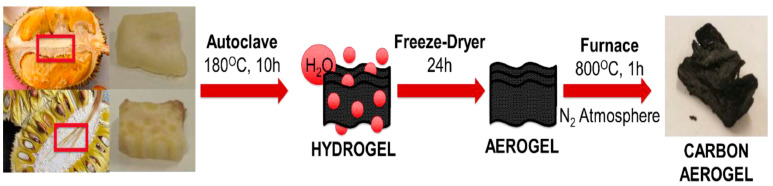
Process of synthesis of aerogel from jackfruit and durian [94].

**Figure 9 polymers-16-02848-f009:**
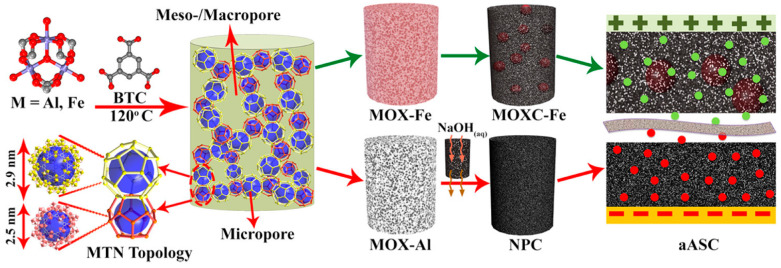
Schematic representation of xerogels derived from MOF [110].

**Figure 10 polymers-16-02848-f010:**
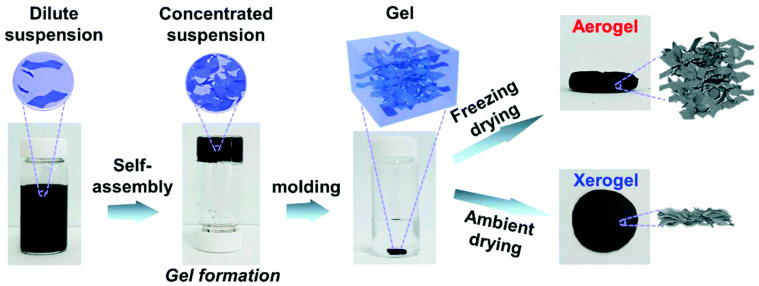
Schematic representation of the synthesis procedure of aerogel and xerogel [117].

**Table 1 polymers-16-02848-t001:** The electrochemical parameters of aerogels and xerogels.

Aerogel/Xerogel	Electrolyte Solution	Specific Capacitance (F/g)/ Current Density (A/g)	Operating Conditions	Power Density/ Energy Density (Wh/kg)	Ref.
Durian carbon aerogel	KOH (0.5 M)	591/1	RT	82.9	[94]
Jackfruit carbon aerogel	KOH (0.5 M)	292/1	RT	40	[94]
NiMoO_4_@Co_3_O_4_/carbon aerogel	KOH (2 M)	436/0.5	RT	208.8	[89]
*Phyllostachys pubescences* nanocellulose graphene aerogel	H_2_SO_4_ (1 M)	125.5	RT	-	[91]
Lignin aerogels	KOH (6 M)	124/0.2	RT	250	[100]
Cellulose nanofibrils/PANI aerogel	H_2_SO_4_ (1 M)	375/0.2	RT	-	[92]
Coal based aerogel	KOH (6 M)	260/1	RT	7.2	[105]
Gelatin based carbon aerogel	KOH (1 M)	236/2	RT	20.88	[29]
CNT embedded carbon xerogel	KOH (6 M)	160/0.1	RT	-	[119]
Nitrogen doped graphene aerogel	KOH (6 M)	325	RT	12.95	[72]
Cellulose based aerogel. MnO_x_	Na_2_SO_4_ (1 M)	269.7	RT	20	[85]
Graphene doped carbon xerogel	H_2_SO_4_ (1 M)	98/16	RT	15	[120]
Activated carbon xerogels	KOH (6 M)	270	RT	-	[31]
Graphene/polyaniline aerogel	H_2_SO_4_ (1 M)	713	RT	6.4	[22]

## Data Availability

Not applicable.

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
