# Peer review of "A Review of Green Aerogel- and Xerogel-Based Electrodes for Supercapacitors"

_polymers, 2024, doi:10.3390/polym16192848_

Round 1
Reviewer 1 Report
Comments and Suggestions for Authors
The search for electrode materials that are much more efficient than the current ones and that provide high energy storage values in electrochemical devices, in accordance with current demand, is a topic of great interest and relevance for the present and the near future. The review work titled "Review on green aerogel and xerogel-based electrodes for supercapacitors" is very interesting and provides a lot of information for the scientific community working in this field.
In my opinion the review is very well organized and written and it is very complete. However, I only have two comments:
- minor comment: supercapacitors are the best complement to batteries for energy storage. They are very stable and can provide a large amount of power, while also being able to recharge very quickly. These characteristics are due to the main phenomena for charge storage in these devices is based in a physical (not chemical) process and therefore it is very fast and reversible. However, the amount of energy storage cannot compete with batteries... Therefore, they can never replace batteries as storage device, but complement them in terms of power. Taking this into account, the authors can correct some idea in the introduction section (lines 38-42), "...superior energy storage capacity" because this is not entirely true and is also in contradiction with the following phrase mentioned by the authors "...supercapacitors are still limited for large-scale application, based on issues such as stability and poor capacitance".
mayor comment: Although the work is very complete, I think there is some discussion and partial conclusions missing in each section. That is, many aspects that are important for the application of sol-gel materials in supercapacitors are mentioned, but the work and results are simply mentioned and what remains to be determined or optimized as future work is not discussed or shown. A brief discussion in each section would be very interesting for the future reader.
On the other hand, in the final conclusions it is worth highlighting that the performance of these materials in supercapacitors also depends on the configuration of the supercapacitor, its size, the electrolyte used and other components of the device, as well as the operating conditions employed. Therefore, the comparison of Table 1 must be done "carefully." Perhaps partial comparisons should be made: aerogels in the same conditions, or aerogel vs xerogel in the same conditions, etc. I recoment to rearrange Table 1 and/or complete it with more information.
Author Response
We are grateful to the Reviewer for a careful and thorough review, and for raising important issues that relate directly to the clarity of the manuscript and the interpretation of the data. After discussing his (her) report, we agree with the suggestion of the Reviewer to submit the revised manuscript to Polymers. We respond to the Reviewer’s comments below and describe the associated revisions to the manuscript.
The search for electrode materials that are much more efficient than the current ones and that provide high energy storage values in electrochemical devices, in accordance with current demand, is a topic of great interest and relevance for the present and the near future. The review work titled "Review on green aerogel and xerogel-based electrodes for supercapacitors" is very interesting and provides a lot of information for the scientific community working in this field.
In my opinion the review is very well organized and written and it is very complete. However, I only have two comments:
- minor comment: supercapacitors are the best complement to batteries for energy storage. They are very stable and can provide a large amount of power, while also being able to recharge very quickly. These characteristics are due to the main phenomena for charge storage in these devices is based in a physical (not chemical) process and therefore it is very fast and reversible. However, the amount of energy storage cannot compete with batteries... Therefore, they can never replace batteries as storage device, but complement them in terms of power. Taking this into account, the authors can correct some idea in the introduction section (lines 38-42), "...superior energy storage capacity" because this is not entirely true and is also in contradiction with the following phrase mentioned by the authors "...supercapacitors are still limited for large-scale application, based on issues such as stability and poor capacitance".
According to the reviewer’s comment, these sentences have been corrected.
mayor comment: Although the work is very complete, I think there is some discussion and partial conclusions missing in each section. That is, many aspects that are important for the application of sol-gel materials in supercapacitors are mentioned, but the work and results are simply mentioned and what remains to be determined or optimized as future work is not discussed or shown. A brief discussion in each section would be very interesting for the future reader.
We thank for the reviewer’s salient observation; We carefully checked the whole manuscript and added the discussion.
On the other hand, in the final conclusions it is worth highlighting that the performance of these materials in supercapacitors also depends on the configuration of the supercapacitor, its size, the electrolyte used and other components of the device, as well as the operating conditions employed. Therefore, the comparison of Table 1 must be done "carefully." Perhaps partial comparisons should be made: aerogels in the same conditions, or aerogel vs xerogel in the same conditions, etc. I recommend rearranging Table 1 and/or complete it with more information.
According to the reviewer’s comment, Conclusions and Table 1 have been changed as suggested.
We sincerely thank the Reviewers for his (her) careful review. We are confident that the detailed responses given above address all of the comments raised by the Reviewers. The revisions and reconsiderations prompted by the Reviewers’ reports have strengthened the manuscript significantly. We have taken the advice of the Reviewers and are submitting the revised manuscript to Polymers.

Reviewer 2 Report
Comments and Suggestions for Authors
The authors present a thorough examination of the synthesis, properties, and potential applications of aerogels and xerogels as electrodes for supercapacitors, demonstrating a commendable level of clarity and organization. However, several areas warrant further elaboration or emphasis:
1. The abstract briefly touches on the use of biomass-derived materials, but a stronger emphasis on the environmental benefits of renewable and sustainable precursors is warranted. Furthermore, the conclusion should underscore the potential for reducing environmental impact through the recycling of organic waste materials.
2. While the conclusion acknowledges the need to scale up synthetic procedures for industrial applications, additional insights into the challenges and potential solutions for transitioning from laboratory to industrial-scale production would enrich the discussion.
3. The importance of flexibility in energy storage devices is crucial. It is recommended that the authors delve deeper into this aspect by discussing the specific mechanical properties required for flexible supercapacitors and how aerogels and xerogels fulfill these criteria.
4. The manuscript outlines the excellent charge-discharge properties of both pristine and hybrid aerogels and xerogels. However, providing more specific details on performance metrics and comparing them to traditional electrode materials would enhance readers' understanding.
5. While the authors touch upon future research directions, more explicit suggestions for areas of investigation or potential technological advancements would provide a clearer roadmap for researchers in the field.
In summary, the manuscript effectively synthesizes key findings and implications regarding green aerogel and xerogel-based electrodes for supercapacitors. Minor revisions to address the aforementioned points would further enhance its clarity and impact. Therefore, I recommend accepting the manuscript after these revisions.
Author Response
We are grateful to the Reviewer for a careful and thorough review, and for raising important issues that relate directly to the clarity of the manuscript and the interpretation of the data. After discussing his (her) report, we agree with the suggestion of the Reviewer to submit the revised manuscript to Polymers. We respond to the Reviewer’s comments below and describe the associated revisions to the manuscript.
The authors present a thorough examination of the synthesis, properties, and potential applications of aerogels and xerogels as electrodes for supercapacitors, demonstrating a commendable level of clarity and organization. However, several areas warrant further elaboration or emphasis:
- The abstract briefly touches on the use of biomass-derived materials, but a stronger emphasis on the environmental benefits of renewable and sustainable precursors is warranted. Furthermore, the conclusion should underscore the potential for reducing environmental impact through the recycling of organic waste materials.
According to the reviewer’s comment, the abstract and conclusion have been corrected.
- While the conclusion acknowledges the need to scale up synthetic procedures for industrial applications, additional insights into the challenges and potential solutions for transitioning from laboratory to industrial-scale production would enrich the discussion.
We thank for the reviewer’s salient observation; We carefully checked the whole manuscript and added the discussion.
- The importance of flexibility in energy storage devices is crucial. It is recommended that the authors delve deeper into this aspect by discussing the specific mechanical properties required for flexible supercapacitors and how aerogels and xerogels fulfill these criteria.
According to the reviewer’s comment, the specific mechanical properties have been updated.
- The manuscript outlines the excellent charge-discharge properties of both pristine and hybrid aerogels and xerogels. However, providing more specific details on performance metrics and comparing them to traditional electrode materials would enhance readers' understanding.
According to the reviewer’s comment, Conclusions and Table 1 have been changed as suggested.
- While the authors touch upon future research directions, more explicit suggestions for areas of investigation or potential technological advancements would provide a clearer roadmap for researchers in the field.
In accord with the reviewer’s comment, the parameters - specific surface area and density has been added.
In summary, the manuscript effectively synthesizes key findings and implications regarding green aerogel and xerogel-based electrodes for supercapacitors. Minor revisions to address the aforementioned points would further enhance its clarity and impact. Therefore, I recommend accepting the manuscript after these revisions.
We sincerely thank the Reviewers for his (her) careful review. We are confident that the detailed responses given above address all of the comments raised by the Reviewers. The revisions and reconsiderations prompted by the Reviewers’ reports have strengthened the manuscript significantly. We have taken the advice of the Reviewers and are submitting the revised manuscript to Polymers.
